# Does Digital Inclusive Finance Development Affect the Agricultural Multifunctionality Extension? Evidence from China

Yafei Wang [1], Jing Liu [1], Huanhuan Huang [1], Zhixiong Tan [2,*] and Lichen Zhang [3]

[1] School of Economics and Management, Chongqing Normal University, Chongqing 401331, China
[2] School of Public Policy and Administration, Chongqing University, Chongqing 400044, China
[3] School of Law, Chongqing University, Chongqing 400044, China
* Correspondence: tzxcqu@126.com

**Abstract:** This paper constructs a comprehensive index system for agricultural multifunctionality extension to measure the agricultural multifunctionality development level in 30 sample provinces in China from 2011 to 2019, builds a model to explain theoretically and test empirically the impact of digital inclusive finance on agricultural multifunctionality extension using Peking University Digital Inclusive Finance Index, and discusses the moderating role of rural human capital in the above process. The main findings include: (1) The rapid increase in the development level of agricultural multifunctionality presents typical regional differences in several regions of China, showing the gradient characteristics of the eastern part higher than the central part and the central part higher than the western part. (2) Digital inclusive finance can significantly promote the agricultural multifunctionality extension, and significantly contribute to the growth of agricultural product supply, economic development and social security functions, while having a particular inhibitory effect on ecological environment function. (3) The width of coverage and depth of use of digital inclusive finance can affect agricultural multifunctionality extension. The digitalization degree inhibits agricultural multifunctionality extension. (4) Rural migratory human, educational human and healthy human capitals are significant positive moderators of the impact of digital inclusive finance on agricultural multifunctionality extension.

**Keywords:** digital inclusive finance; agricultural multifunctionality extension; rural human capital; moderating effect





## 1. Introduction

With the speedy advancement of technology and the steady increase of economic development, people's demands on the functions of agricultural development have changed profoundly, which has driven a historical change of the agricultural development paradigm, from productionism to post-productionism, and to multifunctioning [1,2]. Due to the fact that China has the world's largest population, with relatively little land, China's agricultural development should emphasize the fundamental function of food security as a precondition for diversifying agricultural roles, such as economic growth, social stability and ecological services, which is not only of great practical value for improving the agricultural and rural performance, promoting the growth of farmers' income and achieving rural revitalization, but is also a pragmatic approach to responding to the people's ever-growing needs for a better life [3].

Finance is the lifeline of economic activities, while effective rural financial supply is an indispensable element in agricultural multifunctionality expansion. However, there is financial inhibition and exclusion in rural areas of China. On the supply side, traditional financial institutions have allocated financial resources to the non-agricultural sector in urban areas, and have taken measures, such as closing rural branch offices, in order

to reduce operating costs and maximize profits, owing to the country's long-standing unbalanced development strategy of "industrial priority and urban bias", and financial market-oriented reforms [4]. In the general rural areas, the non-aggregated distribution of rural residents has promoted the scarce construction of financial areas, and coupled with the credit difficulties experienced by rural residents, this leads to higher costs and risks for financial institutions when expanding to the rural market. Moreover, the inherent "weak" characteristics of agriculture [5], such as its long-term production cycles, high natural or market risks, and low returns, also make profit-seeking financial capitals exclude rural areas, leading to a perpetual financial disincentive for rural areas or agriculture [4]. In addition, farmers and modern agricultural business entities have strong financing requirements, which are mainly characterized by the demand for investment in agricultural production and operation, short-term loans for daily consumption, and wealth management at the demand level. Throughout the process of the reallocation of rural land resources and the promotion of agricultural industrialization, farmers have confronted a large credit gap in implementing land transfer and moderate-scale operations. In the background of China's rural revitalization strategy, more and higher-quality requirements for rural finance have been put forward, by the integration of three rural industries, multi-functional expansion of agriculture and rural environmental management, and it is urgent to explore new models and paths of rural financial services. However, the characteristics of rural residents, who are generally less educated, skilled and wealthy, make their repayment abilities relatively low, along with their lack of sufficient and readily realizable collateral, which causes the "elite capture" orientation in the selection of credit allocation targets and the exclusion of the majority of low- and middle-income rural households from formal financial services.

Inclusive finance aims to provide appropriate and effective financial services at an affordable cost to all social classes and groups in need of financial services, based on equal opportunity requirements and commercial sustainability principles [6]. The inherent "market deficiencies" in the development of traditional rural finance provide a profitable space for inclusive finance to target rural market segments that were previously "excluded" by traditional finance. With the increased development of mobile Internet, big data, artificial intelligence and other digital technologies, a digital-driven financial inclusion model is being formed at an accelerated pace around the world. It provides a wide range of financial services, such as payments, savings and wealth management, to more people, especially those who originally faced "financial exclusion", as well as ensures that everyone with access to the Internet is able to obtain financial services at a low cost. As digital infrastructure and mobile Internet technology penetrate widely into rural areas and agricultural fields, the digital environment in rural areas is becoming more and more complete, and application scenarios are becoming more and more abundant. Providers are actively expanding their market space to rural areas, including areas dedicated to agriculture and farmers, and are introducing differentiated and inclusive financial products or services for agricultural production and business entities, as well as the general rural inclusive community.

At the practical level, digital inclusive finance has been developing rapidly in rural areas, forming a rich variety of rural digital inclusive finance development models. One is the financial institution-based rural digital inclusive finance model. With the support of mobile Internet, big data and intelligent technologies and tools, traditional financial institutions have expanded digital, inclusive finance services to rural business entities through online banking, mobile banking and e-commerce platforms, in order to satisfy the need for diversified financial products or innovative service needs, such as Construction Bank's "Yu Nong Tong", Agricultural Bank's "Huinong e Tong", Ningxia Bank's "Ruyi Support Loan" and "Ruyi Farming Loan", and the "Villager e-Loan" of Guangzhou Agricultural Bank. The second is the agricultural supply chain financial service provider-based, rural, digital, inclusive finance model. As the main organizer or guide in the agricultural supply chain, agricultural leading enterprises systematically integrate the commercial flow, information flow, capital flow, financing demand and credit risk data of related business entities, in order to provide more accurate and convenient digital, inclusive financial products or

services in the supply chain, with the help of big data and artificial intelligence technology. For example, the "Nongfudai", "Nongnongdai" and "supporting fund" of Dabinong, the "village loan" and "village finance" of New Hope, and the "Huinongdai", "hope treasure" and "loan should be collected" of the village music. The third is the financial technology company-based, rural, digital, inclusive finance model. Some large, integrated e-commerce platforms, such as Alibaba, Jingdong, Tencent and other financial technology companies, have given full play to their comprehensive, comparative advantages, such as their wide user coverage and use of big data integration to penetrate a wide range of flexible digital inclusive financial products or services to rural target groups, in order to facilitate or alleviate the financing bottlenecks of relevant agricultural business entities or farmers. Examples of these include Alibaba's "Wangnong Loan", "Wangnong Insurance" and "Wangnong Pay", Jingdong's "Rural White Strip" and "Jingnong Loan", and Wanhui Crowdfunding's "Yinong Loan".

Extensive studies have confirmed that digital inclusive finance has contributed to the convergence of the urban–rural earnings and enhanced agriculture production and operation efficiency. However, the impact of digital inclusive finance on agricultural multifunctionality extension has often been ignored. In fact, "agricultural multifunctionality" has been repeatedly mentioned in policy documents and academic research since the Chinese government first explicitly stated that "multiple functions of agriculture should be developed" in 2007. The 14th Five-Year Plan of China has made it clear that agricultural multifunctionality extension is the essential substance and path that will be taken for the strategy of promoting rural industries' integration, and enriching the rural economy and business. Thus, the inclusion of digital inclusive finance into the agricultural multifunctionality extension framework, and the empirical examination of digital inclusive finance's influence on agricultural multifunctionality extension will not only expand the theoretical perspectives in the efficiency assessment of digital inclusive finance, as well as provide insights into the impact and problems, but will also provide a rich policy implication for rural digital inclusive finance and high-quality agriculture.

In addition, rural human capital is also an essential factor in digital inclusive finance, supporting the functional development of agriculture. A high-quality labor force is indispensable to support agricultural multifunctionality, economic development, social security and ecological services. Meanwhile, the level of rural human capital is crucial for the awareness and recognition of, and eventual access to, digital inclusive finance. Agricultural business subjects at different human capital levels differ in their internet knowledge, financial literacy and capital allocation ability, so there are inevitable differences in the their efficiencies. There are inevitable differences in the efficiencies of production and operation activities after making digital inclusive finance financing decisions and acquiring financial resources. Therefore, human capital can have a substantial impact on the performance of digital inclusive finance for agricultural multifunctionality extension, and the discussion of this issue contains a wealth of policy implications.

The main contributions include constructing an agricultural multifunctionality evaluation index system, taking the connotation of agricultural multifunctionality as the basis, by collecting data from 30 provincial samples in China from 2011 to 2019, applying the entropy weight method to measure the development level of agricultural multifunctionality, and examining the evolutionary dynamics and regional differences in agricultural multifunctionality, in terms of the dual dimensions of time evolution and spatial differences. Secondly, we incorporate digital inclusive finance into the analytical framework of agricultural multifunctionality and empirically examine the impact and effect of digital inclusive finance on the development of agricultural multifunctionality, as well as the moderating role played by rural human capital investments, such as healthiness, education and training, and migration, in the context of the above impact, thus enriching or extending the research findings surrounding the driving mechanism of agricultural multifunctionality development. At the theoretical level, this study helps to reveal the intrinsic link between digital inclusive finance and agricultural multifunctionality development; at the practical

level, it conducts a long-term, multi-regional-level evaluation of agricultural multifunctionality, and an in-depth analysis of its temporal evolution and spatial differences, which will help in gaining a comprehensive understanding of the effectiveness of agricultural multifunctionality development in China, as well as the unevenness in it between regions. It provides a method and approach reference for other countries to evaluate the level and spatial variation of agricultural multifunctionality. Furthermore, this study confirms digital inclusive finance's role in facilitating agricultural multifunctionality, as well as the role of different types of rural human capital investment, providing a wealth of policy insights for the promotion of agricultural multifunctionality and sustainable agricultural growth, through digital inclusive finance development and rural human capital investment globally, especially in developing countries.

## 2. Theoretical Analysis

### 2.1. The Impact of Digital Inclusive Finance on the Agricultural Multifunctionality Extension

Agricultural multifunctionality covers multiple functional dimensions, such as agricultural product supply, economic development, social security and ecological services. It comprehensively reflects the function and role of each subdivisional function and the interactions between them. Digital inclusive finance's impact on agricultural functions depends on the magnitude and direction of each subdivision's impact. Therefore, this study focuses on each segmented function, and tries to explain the effect of the mechanism of digital inclusive finance's effect on agricultural multifunctionality extension.

Agricultural product supply function: in the application of traditional finance to agricultural fields, the phenomenon of "financial exclusion" and "elite capture" appeared, while digital inclusive finance can decrease transaction costs, relieve financial risks, and expand customer service targets, which are generated by the organic combination of internet, digital technology and finance [7]. The wide penetration of digital inclusive financial resources to agricultural production and management subjects, providing them with low-cost, facilitated and flexible financial resources, can promote the expansion and reproduction of agricultural production and management activities, which, in turn, support the enhancement of agricultural supply capacity [8]. With the effective supply of digital inclusive finance, agricultural production operators can enhance the scale and efficiency of agricultural supply through the systematic integration of financial resources and production factors, such as labor, land, and advanced agricultural technology and management, in order to promote the technical level of agricultural production and optimize the factor allocation. In addition, the enhancement of the supply function of agricultural products depends not only on the optimization and capacity enhancement of the agricultural production system itself, but also on the expansion of agricultural market demand. Numerous financial institutions have implemented a financial business model for the supply chain of agricultural products that integrates digital inclusive finance with logistics, business flow, and information flow, enhancing the overall efficiency of the supply chain and expanding the sale radius of agricultural products [9]. Additionally, digital inclusive finance also drives the expansion of agricultural market demand through the consumption scale enhancement and consumption structure upgrading effects on urban residents [10], which, in turn, promotes the enhancement of agricultural supply function at the external demand level.

Economic development function: the proportional decline in the contributions of agriculture to the national economy is an important indication of the law of industrial evolution, and digital inclusive finance cannot fundamentally reverse this trend. However, similar to the previous analysis, digital inclusive finance can improve the overall operation efficiency of agricultural production system through the support effect of financial factors and the demand-induced effect. As the production and operation activities of agricultural products themselves are an important part and driving force of the agricultural economic operation system, the enhancement effect of digital inclusive finance on the supply function of agricultural products will also affect the growth of agricultural economy. Moreover, many studies have verified that digital inclusive finance, compared with the inherent



shortcomings of traditional finance, can promote agricultural economic growth and internal structure optimization, as well as display a strong agricultural input–output efficiency enhancement effect [11].

Social security function: the social security function of agriculture is mainly reflected in rural employment and farmers' income growth. First, digital inclusive finance helps farmers' to expand their scale, by providing them with effective financial support to engage in production and business activities, such as expanded reproduction and moderate-scale operations, with financial support, which leads to the increase in output and the improvement of factor allocation efficiency, which consequently leads to the increase in farmers' incomes [12]. Furthermore, the production efficiency improvement of agriculture itself promotes, to a certain extent, the non-farm employment of surplus labor from farming households, which increases the wage income of farming households or families. Second, the spillover effect of knowledge and technology embedded in digital inclusive finance can enhance the human capital level in rural areas [13] and thus trigger the growth of farmers' income and the improvement of rural human capital quality driven, by quality of life improvements. Furthermore, digital inclusion finance promotes rural residents' entrepreneurial behavior, especially among rural low-income and low-social-capital households [14]. The intrinsic mechanism of this is primarily manifested in the easing of credit and information constraints and the strengthening of social credit [15].

Ecological service function: the ecological function of agriculture is displayed as the contribution to environmental quality and ecology, etc. Digital inclusive finance is characterized by highly green attributes. Its networked, digitalized and intensive business model can successfully lower energy usage and pollution in financial services, produce good social demonstration effects, and enhance the ecological and environmental awareness of relevant entities [16]. By embedding digital technologies such as big data, artificial intelligence, blockchain, etc., digital inclusive finance helps to form an environmental information [17] disclosure and sharing mechanism, as well as a project selection, environmental supervision and fund allocation mechanism, based on environmental information [17], which helps to reduce the moral hazards and opportunistic behavior, regarding agricultural production and management entities and related participants who destroy the environment in pursuit of profit maximization. This further reduces the damage to agricultural ecological environment, and improves the efficiency of rural or agricultural ecological governance.

Based on the interpretation of the above four segmented functions, this study proposes four hypotheses.

**Hypothesis 1.** *Digital inclusive finance development contributes to the strengthening of the function of agricultural product supply.*

**Hypothesis 2.** *Digital inclusive financial development supports the enhancement of the economic development function of agriculture.*

**Hypothesis 3.** *Digital inclusive financial development promotes the social security function of agriculture.*

**Hypothesis 4.** *Digital inclusive financial development helps enhance the ecological service function of agriculture.*

*2.2. The Moderating Effect of Rural Human Capital*

Earlier, we explained the mechanism of how digital inclusive finance impacts agricultural multifunctionality. Is there any other factor that will further strengthen or inhibit agricultural multifunctionality extension by acting in conjunction, or in synergy, with digital inclusive finance? As mentioned earlier, the inclusive and digital characteristics of digital inclusive finance will improve the financial supply conditions of agricultural development, and ease the "financial constraints" in the process of agricultural multifunctionality exten-

sion. Furthermore, the accompanying penetration and integration of digital technology into agriculture will improve the operation level or governance efficiency of the agricultural industry, and provide digital technology or management support for agricultural multifunctionality extension. It should not be overlooked, however, that human capital is the most active or creative of the production factors in modern industrial transformation, and that the conditions for a human capital endowment to match the evolution of industry are still absent or lacking. Even if the supply conditions of other factors improve, the speed and performance of industrial transformation or industrial evolution will be significantly reduced. Therefore, it is necessary to include rural human capital in the discussion of the driving mechanism of agricultural multifunctionality development, in order to better explain the logical motivation of China's agricultural multifunctionality development and put forward richer policy implications.

The formation of human capital largely depends on the investment level of rural human capital, which mainly includes three aspects. One is healthy human capital, based on medical insurance expenditure. Human capital investments that result in improvements in the health of households or populations can improve long-term business performance by increasing investment confidence, as well as financial and other factors' allocation ability [18,19]. The second aspect is the education–training human capital, based on education and training expenditure. Education and training expenditures promote the knowledge and skill level of labor factors, enhance the market-oriented cognitive and operational capabilities, and improve the factor allocation efficiency of industries and market operators to a certain extent, which is important in upgrading the innovation of industrial formats and business models [20]. The third aspect is migrant human capital, based on transportation and communication expenditure. Migrant human capital investment broadens the geographical flow boundary of investors, expands interpersonal networks, contributes to access to newer business information, knowledge and experience and social capital, and has a critical leading effect on innovation and entrepreneurship [21].

In addition, with the digital inclusive finance supply, the empowering effect of rural human capital investment on rural or agricultural entities will realize the organic combination of financial capital, digital technology and high-quality labor factors, improve the comprehensive allocation efficiency of various agricultural production factors, and lead to the optimal adjustment of agricultural industrial systems, as well as production and operation systems, and thus increase the level of agricultural multifunctionality. Therefore, this study proposes Hypothesis 5:

**Hypothesis 5.** *Rural health human capital, education and training human capital and migration human capital all play a positive moderating role in the impact of digital inclusive finance on agricultural multifunctionality extension.*

### 3. Models, Estimation Methods and Variables

*3.1. The Connotation of Agricultural Multifunctionality and the Construction of an Index System*

In the context of history, the connotation of agricultural multifunctionality can be traced back to the "rice culture" proposed by Japan around the 1990s. In global terms, practical and theoretical circles define agricultural multifunctionality from the perspective of functional attributes. The OECD, FAO, EU and WTO generally stated that the agricultural multifunctionality mainly involves the product supply capacity, and economic, social, cultural and ecological function from the perspective of "sustainable" agricultural development. This pioneering formulation has provided a relatively unified reference system for later interpretations of agricultural multifunctionality's connotation and evaluation direction [22]. Some studies have deepened or expanded the connotation of agricultural multifunctionality in China. However, most studies have approached the issue of agricultural subdivisions from the perspective of "food security", based on the actual national dynamics of developing countries with the largest populations, which further highlights the fundamental position of agricultural supply function in many subdivisions.

Agricultural academia has continuously focused on the formulation of a multivariate index system for evaluating the development level of multifunctionality and the segmented functions of the agricultural sector in recent years. Perhaps because of the complexity of constructing an evaluation system of agricultural multifunctionality or the high difficulty in terms of data collection, scholars are more likely to conduct situational or comparative studies from specific provinces or cities, and rarely conduct them at the national, regional and provincial levels in time and space. This study argues that measuring and analyzing the spatial and temporal evolution of China's agricultural multifunctionality is an invaluable clue to understanding and interpreting the current "unbalanced and insufficient" contradiction in China. Therefore, based on the mainstream view of agricultural multifunctionality, this study constructs an evaluation index system that includes agricultural product supply, economic development, social security and ecological service functions. In fact, the cultural inheritance function is also an important branch of agricultural multifunctionality, but the indicators, especially the data, of this function are difficult to obtain from the macro data of official statistics, so this study does not include it. The evaluation indexes and measurement methods of agricultural multifunctionality and some literature references are in Table 1.

**Table 1.** Index system of agricultural multifunctionality evaluation.

| Decision-Making | Target | Indicator | Forward/Reverse | Indicator Calculation Method | Indicator Source |
|---|---|---|---|---|---|
| Agricultural multifunctional evaluation | Agricultural product supply function | Food self-sufficiency rate | + | Food production/total regional population—400 kg/person | — |
| | | Grain yield | + | Grain production/grain sown area | [23] |
| | | Vegetable production per capita | + | Vegetable production/total regional population | — |
| | | Fruit production per capita | + | Fruit production/total regional population | — |
| | | Oilseed crop production per capita | + | Oilseed crop production/total regional population | — |
| | | Meat, egg and milk production per capita | + | Meat, egg and milk production/total regional population | — |
| | | Aquatic production per capita | + | Aquatic production/total regional population | — |
| | | Percentage of effective irrigated area | + | Effective irrigated area/cultivated land area | [24] |
| | | Primary industry value-added ratio | + | Value added of primary production/GDP | [25] |
| | Agricultural product supply function | Value added per capita of the primary industry | + | Value added of primary production/total regional population | — |
| | | Value added per capita of forestry, livestock and fisheries | + | Forestry, livestock and fishery added value/total regional population | [26] |
| | | Reproduction index | + | Area of food crops sown/area under cultivation | — |
| | | Rural employment rate | + | Rural employed population/total rural population | — |
| | Agricultural product supply function | Regional average total mechanical power | − | Total power of agricultural machinery/arable land area | — |
| | | Average labor value-added of the primary industry | + | Value added of primary production/rural employment | — |
| | | Cultivated land per capita | + | Arable land area/total rural population | [27] |
| | | Agriculture, forestry and water affairs expenditure per capita | + | Agriculture, forestry and water affairs expenditure/total rural population | — |

**Table 1.** *Cont.*

| Decision-Making | Target | Indicator | Forward/Reverse | Indicator Calculation Method | Indicator Source |
|---|---|---|---|---|---|
| Agricultural multifunctional evaluation | Agricultural product supply function | Forest coverage rate | + | Forest area/administrative area | [28] |
| | | Farmland Ecosystem Diversity Index | + | Sown area of each crop/total sown area | — |
| | | NDVI mean value | + | Regional annual NDVI values | — |
| | | Fertilizer application load | − | Fertilizer application/arable land area | [27] |
| | | Pesticide application load | − | Pesticide application/arable land area | [29] |

### 3.2. Econometric Model and Estimation Method

Based on Difference GMM, the improved System GMM method can significantly alleviate the model's endogeneity and improve the robustness of parameter estimation [30,31]. Systematic GMM estimation is further divided into one-step and two-step estimation methods, on the basis of the difference in weight model selection. In the usual case, the standard covariance matrix of the two-step estimation method can deal with serial autocorrelation and heteroskedasticity more effectively, with a more robust estimation [32,33]. Assuming the dynamic persistence of agricultural multifunctionality, this study uses agricultural multifunctionality lagged by one period as the explanatory variables, and estimates the impact of digital inclusive finance on agricultural multifunctionality extension using a two-step dynamic system GMM model to control some omitted factors and biases more effectively; the model is constructed as follows:

$$AV_{it} = \alpha + \beta_1 LNDFI + \beta_2 AV_{it-1} + \beta_3 X_{it} + \mu_i + \lambda_t + \varepsilon_{it} \tag{1}$$

In Formula (1), *DFI* is the core explanatory variable, representing the degree of digital financial inclusion; *AV* is the explained variable, representing agricultural multifunctionality; *X* is the set of control variables, *LN* represents the natural logarithm, *i* and *t* represents the *i*th province and *t*th period respectively, $t-1$ represents the lag phase, $\mu_i$ represents the fixed effect of region, $\lambda_t$ represents the fixed effect of year, and $\varepsilon_{it}$ represents the random disturbance term.

In addition, to investigate the moderating effect of rural human capital on the digital inclusive finance's impact on the development level of agricultural multifunctionality, we further added the interaction term between digital inclusive finance and rural human capital into the benchmark model (1), and obtained model (2):

$$AV_{it} = \alpha + \beta_1 LNDFI_{it} + \beta_2 AV_{it-1} + \beta_3 LN(DFI_{it} \times HC_{it}) + \beta_4 X_{it} + \mu_i + \lambda_t + \varepsilon_{it} \tag{2}$$

In model (2), $DFI_{it} \times HC_{it}$ represents the interaction term between digital inclusive finance and rural human capital, which is specifically subdivided into migratory human capital, educational human capital and health human capital, and $X_{it}$ represents control variables. To simplify the model and to avoid an inability of the model to be identified, due to the addition of too many parameters, the model only contains one interaction item during the inspection process, which is checked successively. For parameter estimation of the model (1) and model (2), we adopt the two-step, dynamic System GMM method.

### 3.3. Variable Definition

Explanatory variables: agricultural multifunctionality (*AV*) is measured using the comprehensive index of agricultural multifunctionality, as measured above. The higher the comprehensive index value is, the higher the development level of agricultural multifunctionality is. The agricultural multifunctionality index is also subdivided into the agricultural product supply function (*PSL*), agricultural economic development function (*EDL*), agricultural social security function (*SSL*), and agricultural ecological service function (*EEL*).

Core explanatory variables: we selected the Digital Inclusive Finance Development Index of China for provinces and cities, from 2011 to 2019, as compiled by Guo Feng [34] as a proxy variable for the digital inclusive finance development (*LNDFI*) level. Digital inclusive finance is separated into coverage breadth (*LNCOV*), usage depth (*LNUSE*), and digitization degree (*LNDIG*). The index, jointly compiled by Peking University and Ant Financial, is based on massive real transaction data of users and is authoritative, scientific and reasonable to a certain extent. To reduce the difference among the orders of magnitude of each variable, the above variables are logarithmically treated.

Control variables: to minimize the bias resulting from omitted variables, we selected some of the following variables as control variables in this study, on the foundation of existing research. Urbanization (*LNURBAN*) is characterized by the share of the urban population in the total population at the year's end; government fiscal expenditure (*LNFE*) is measured by the share of government fiscal expenditure in the total regional value; foreign direct investment (*FDI*) is expressed by the share of *FDI* in the regional GDP of each province. The industrial structure (*LNTG*) is defined by the number of tertiary industries relative to the regional GDP in each province. Economic development (*LNGDP*) is embodied by constant-price GDP, with2011 as the base period. To mitigate heteroskedasticity and reduce differences among the magnitude orders of each variable, urbanization, government fiscal expenditure, industrial structure, and economic development are all log-adjusted.

Adjusting variables: for the rural human capital indicator (*HC*), this study uses transportation and communication expenditures (*MH*) to express migratory human capital, cultural, educational, and recreational expenditures (*EH*) to represent educational human capital, and health care expenditures (*HH)* to characterize health human capital. The details are described as follows.

Migratory human capital: the migration of rural human capital contributes to agricultural diversity and regional innovation. Increasing the investment in agricultural transportation and communication will effectively strengthen the inter-regional interaction among rural residents and improve the spread of financial knowledge and agricultural diversification regionally [35]. We use the ratio of transportation and communication expenditure to total expenditure to represent the migratory human capital.

Educational human capita: under the dualistic urban–rural structure, the comprehensive quality and education of rural residents have been the focus of attention at home and abroad. It is of great significance to improve farmers' basic quality of life and skills in the current agricultural diversification and modernization process. A high level of human capital not only enables farmers to master financial knowledge and improve their financial state, but also indirectly improves the development of the agricultural multifunctionality level [36]. This paper uses the culture, education and entertainment expenditure as a proportion of the total expenditure to represent educational human capital.

Healthy human capital: to improve agricultural production efficiency and to contribute to agricultural diversity quality, it is essential to invest in human health care and increase infrastructure construction, in order to ensure farmers' health security [37]. In this paper, the amount of healthcare expenditure as a proportion of the total expenditure is used to represent healthy human capital. The selection of variables and their meanings are in Table 2.

**Table 2.** Variable description.

| Names of Variables | Code | Variable Declaration |
|---|---|---|
| Agricultural Multifunctionality Development | AV | Agricultural multifunctionality measured by entropy method |
| Digital Inclusive Finance | LNDFI | Peking University Digital Inclusive Finance Index |
| Urbanization Rate | LNURBAN | Share of the urban resident population in total population (%) |
| Government Fiscal Spending | LNFE | Share of government fiscal expenditure in GDP by the province in the current year (%) |
| Foreign Direct Investment | FDI | The amount of actual utilization of foreign direct investment as a proportion of GDP (%) |
| Industrial Structure | LNTG | Share of tertiary industry output value in GDP by province (%) |
| Economic Development | LNGDP | GDP per capita in constant prices with 2011 as the base period |
| Breadth of Digital Inclusive Financial Coverage | LNCOV | Decomposition of the Digital Inclusive Finance Index of Peking University |
| Depth of Use of Digital Inclusive Amount | LNUSE | Decomposition of the Digital Inclusive Finance Index of Peking University |
| Digital Inclusive Amount Digitization | LNDIG | Decomposition of Digital Inclusive Finance Index of Peking University |
| Agricultural Product Supply Function | PSL | Agricultural Multifunctionality Segmentation Function |
| Agricultural economic development function | EDL | Agricultural Multifunctionality Segmentation Function |
| Agricultural social security function | SSL | Agricultural Multifunctionality Segmentation Function |
| Agricultural ecological environment function | EEL | Agricultural Multifunctionality Segmentation Function |
| Migratory human capital | MH | Share of transportation and communication expenditures in total consumption expenditures (%) |
| Educational human capital | EH | Share of expenditure on culture, education and entertainment in total consumption expenditure (%) |
| Healthy human capital | HH | Share of medical and health care expenditures in total consumption expenditures (%) |

*3.4. Data Sources and Descriptive Statistics*

This study constructed a comprehensive index system to obtain the development level of agricultural multifunctionality, using the panel data of 30 provinces in China, from 2011 to 2019. Digital inclusive finance, calculated by Peking University, was used as the basic data, and the data of the control variables came from China's Statistical Yearbook and The National Bureau of Data Statistics, covering a period from 2011 to 2019. Descriptive statistics are shown in Table 3.

**Table 3.** Descriptive statistics of variables.

| Variable | Sample Size | Mean | SE | Max | Min |
|---|---|---|---|---|---|
| AV | 270 | 0.28 | 0.06 | 0.44 | 0.16 |
| DFI | 270 | 203.36 | 91.57 | 410.28 | 0.32 |
| URBAN | 270 | 57.63 | 12.18 | 89.61 | 34.97 |
| FE | 270 | 24.94 | 10.32 | 63.37 | 11.03 |
| FDI | 270 | 1.98 | 1.51 | 7.96 | 0.01 |
| TG | 270 | 48.63 | 8.97 | 87,205.38 | 32.66 |
| GDP | 270 | 21,500.02 | 17,134.13 | 1471.97 | 69.59 |
| PSL | 270 | 0.12 | 0.04 | 0.24 | 0.05 |
| EDL | 270 | 0.04 | 0.02 | 0.08 | 0.00 |
| SSL | 270 | 0.07 | 0.05 | 0.26 | 0.02 |
| EEL | 270 | 0.05 | 0.01 | 0.08 | 0.02 |
| COV | 270 | 183.61 | 90.24 | 384.66 | 1.96 |
| USE | 270 | 197.97 | 91.35 | 439.91 | 6.76 |
| DIG | 270 | 278.38 | 118.00 | 462.23 | 7.58 |
| MH | 270 | 12.55 | 2.44 | 17.90 | 8.10 |
| EH | 270 | 9.58 | 2.49 | 14.8 | 4.50 |
| HH | 270 | 9.57 | 2.12 | 16.80 | 4.90 |

## 4. Results and Discussion

### 4.1. The Evolution Trend

This study is based on panel data regarding agricultural multifunctionality extension in 30 provinces of mainland China, excluding Hong Kong, Macao, Taiwan and Tibet, from 2011 to 2019, and is measured by the entropy method. Figure 1 presents the spatio-temporal evolution of agricultural multifunctionality extension from 2011 to 2019 across the country; Table 4 shows the inter-regional evolution of agricultural multifunctionality extension and its subdivisional functionality in the eastern, central and western regions, and their respective provinces.

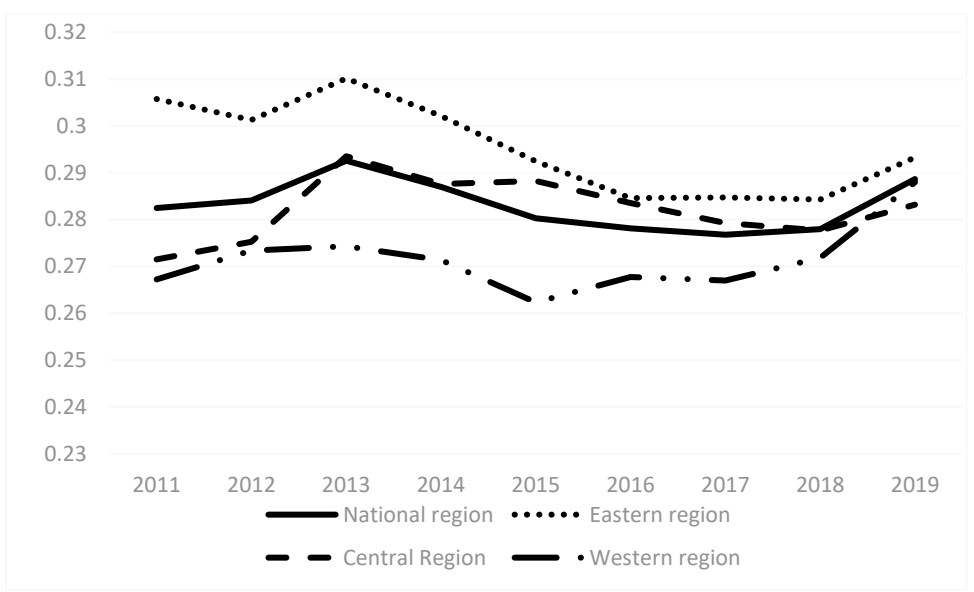

**Figure 1.** Evolution trend of agricultural multifunctionality extension in China from 2011 to 2019.

Figure 1 reports the trend of China's agricultural multifunctionality evaluation index from 2011 to 2019. The agricultural multifunctionality index increased from 0.282 in 2011 to 0.289 in 2019, roughly experiencing three intervals, including an upward cycle from 2011 to 2013, a downward cycle from 2013 to 2017, and an upward cycle from 2017 to 2019. Among regions, the western region maintained a steady growth trend, and its amplitude was the most noticeable. The central region showed a slight trend of growth, while the eastern region showed a trend of decline.

In order to further characterize the development level of China's agricultural multifunctionality and each segmented function at the provincial level, we measured the mean values of agricultural multifunctionality and each segmented index of 30 sample provinces from 2011 to 2019, as shown in Table 4. In terms of overall functionality, Hainan has the highest development level (0.42), followed by Inner Mongolia (0.39), Heilongjiang (0.38), Shanghai (0.35) and Xinjiang (0.34). Tianjin has the lowest development level (0.17). Shanxi (0.18), Gansu (0.21), Qinghai (0.21) and Guizhou (0.22) followed.

In terms of the various subdivisions of agriculture, the highest development level of agricultural supply function is in Hainan (0.23), followed by Shandong (0.20), Fujian (0.19), Henan (0.18), Xinjiang (0.18), etc., and the lowest is in Shanxi (0.06), followed by Beijing (0.07), Tianjin (0.07), Shanghai (0.07), Guizhou (0.07), etc. In terms of the economic development function, Hainan has the highest level (0.08), followed by Anhui (0.06), Jiangxi (0.06), Henan (0.06), Hunan (0.06), and Beijing has the lowest level (0.01); Tianjin (0.02), Shanghai (0.02), Zhejiang (0.02) and Shanxi (0.02) follow. In terms of the social security function, Shanghai has the highest development level (0.23), followed by Beijing (0.17), Heilongjiang (0.17), Inner Mongolia (0.13) and Xinjiang (0.11). Hunan has the lowest (0.02), followed by Hebei (0.03), Fujian (0.03), Shandong (0.03) and Guangdong (0.03). Guizhou

(0.08) and Yunnan (0.08) has the highest level of ecological service function, followed by Zhejiang (0.07), Heilongjiang (0.07) and Guangxi (0.07).

**Table 4.** Average values of agricultural multifunctionality, and subdivision indices of 30 sample provinces in China from 2011 to 2019.

| | Provinces | Overall Agricultural Multifunctionality | Agricultural Product Supply Function | Economic Development Function | Social Security Function | Ecological Service Function |
|---|---|---|---|---|---|---|
| | Beijing | 0.31 | 0.07 | 0.01 | 0.17 | 0.06 |
| | Tianjin | 0.17 | 0.07 | 0.02 | 0.05 | 0.04 |
| | Hebei | 0.26 | 0.14 | 0.05 | 0.03 | 0.05 |
| | Liaoning | 0.32 | 0.15 | 0.05 | 0.06 | 0.06 |
| | Shanghai | 0.35 | 0.07 | 0.02 | 0.23 | 0.03 |
| Eastern region | Jiangsu | 0.28 | 0.15 | 0.05 | 0.05 | 0.04 |
| | Zhejiang | 0.27 | 0.14 | 0.02 | 0.05 | 0.07 |
| | Fujian | 0.32 | 0.19 | 0.04 | 0.03 | 0.05 |
| | Shandong | 0.33 | 0.20 | 0.05 | 0.03 | 0.04 |
| | Guangdong | 0.22 | 0.12 | 0.03 | 0.03 | 0.05 |
| | Hainan | 0.42 | 0.23 | 0.08 | 0.06 | 0.05 |
| | Mean | 0.30 | 0.14 | 0.04 | 0.07 | 0.05 |
| | Shanxi | 0.18 | 0.06 | 0.02 | 0.04 | 0.05 |
| | Jilin | 0.28 | 0.09 | 0.05 | 0.08 | 0.06 |
| | Heilongjiang | 0.38 | 0.09 | 0.05 | 0.17 | 0.07 |
| Central region | Anhui | 0.26 | 0.13 | 0.06 | 0.03 | 0.05 |
| | Jiangxi | 0.26 | 0.12 | 0.06 | 0.03 | 0.06 |
| | Henan | 0.30 | 0.17 | 0.06 | 0.03 | 0.04 |
| | Hubei | 0.32 | 0.16 | 0.05 | 0.04 | 0.06 |
| | Hunan | 0.27 | 0.13 | 0.06 | 0.02 | 0.06 |
| | Mean | 0.28 | 0.12 | 0.05 | 0.06 | 0.06 |
| | Inner Mongolia | 0.39 | 0.15 | 0.05 | 0.13 | 0.06 |
| | Guangxi | 0.29 | 0.14 | 0.05 | 0.03 | 0.07 |
| | Chongqing | 0.24 | 0.08 | 0.03 | 0.05 | 0.07 |
| | Sichuan | 0.28 | 0.11 | 0.05 | 0.05 | 0.07 |
| | Guizhou | 0.22 | 0.07 | 0.03 | 0.04 | 0.08 |
| Western region | Yunnan | 0.24 | 0.07 | 0.04 | 0.05 | 0.08 |
| | Shaanxi | 0.26 | 0.12 | 0.03 | 0.05 | 0.07 |
| | Gansu | 0.21 | 0.08 | 0.02 | 0.06 | 0.05 |
| | Qinghai | 0.21 | 0.07 | 0.03 | 0.07 | 0.05 |
| | Ningxia | 0.29 | 0.14 | 0.03 | 0.08 | 0.05 |
| | Xinjiang | 0.34 | 0.18 | 0.04 | 0.11 | 0.03 |
| | Mean | 0.27 | 0.11 | 0.04 | 0.07 | 0.06 |
| National region | Mean | 0.27 | 0.13 | 0.03 | 0.06 | 0.05 |

In addition, at the level of the three regions, the overall function of agriculture shows a gradient characteristic, with the eastern region higher than the central region and the central region higher than the western region, which indicates that the agricultural overall function in a region is still relatively consistent with its economic development level. At the level of each subdivision of agricultural multifunctionality, the functions of agricultural products supply from high to low are the eastern, central and western regions; the functions of economic development of the eastern region are the same as that of the western region and lower than that of the central region, and the social security function of the eastern region is the same as that of the western region and higher than that of the central region. The ecological service function of central China is the same as that of western China and higher than that of eastern China. When we further observe the role of each sub-function in the overall function, the agricultural supply function has an absolute advantage in each sub-

function, except for Beijing and Heilongjiang, where the social security function is dominant, and Qinghai, where the social security function and the agricultural supply capacity are roughly equal. This implies that China's agricultural multifunctionality expansion has effectively responded to the national demand for agricultural or food security, and such a development is of great practical value in enhancing the strategic position of agriculture in the national economy, and building a new development pattern of agricultural "dual circulation".

Among the subdivision functions of agriculture, the economic development function is the lowest (0.4), which is significantly lower than the other segmented functions, indicating that the decline of agriculture's role in national economic growth with the accelerated urbanization and industrialization, which is broadly in line with the general rule of China's transition from an agricultural country to an industrial country and industrial structure upgrading. However, in the context of relative poverty management and rural revitalization, it is still necessary to make great efforts to explore the potential of enhancing the economic value of agriculture to highlight better the role of agricultural industry revitalization in rural revitalization.

### 4.2. Results of Two-Step System GMM

First, this study uses a two-step dynamic system GMM method to estimate the parameters of Equation (1), and the estimation results are shown in Table 5, including model I–IV. Model I is the estimation result without controlling for area and time effects. Model II is the estimation result without controlling for area effects but controlling for time effects. Model III is the estimation result without controlling for time effects but controlling for area effects, and Model IV is the estimation result controlling for both area and time effects. This study focuses on Model IV as the baseline model for discussion.

**Table 5.** Estimated results of the impact of digital financial inclusion on overall agricultural multifunctionality.

| | Model I (1) | Model II (2) | Model III (3) | Model IV (4) |
|---|---|---|---|---|
| $AV_{it-1}$ | 0.614 *** | 0.663 *** | 0.453 *** | 0.740 *** |
| | (17.10) | (9.64) | (6.25) | (11.45) |
| LNDFI | 0.00403 * | 0.0638 *** | 0.00878 ** | 0.0754 *** |
| | (1.76) | (2.86) | (2.08) | (3.49) |
| LNURBAN | −0.0270 | −0.0229 | 0.165 *** | −0.0636 *** |
| | (−1.39) | (−0.61) | (7.70) | (−3.48) |
| LNFE | −0.00939 | −0.0110 | 0.0602 *** | −0.0232 |
| | (−0.91) | (−0.44) | (5.03) | (−1.19) |
| FDI | 0.00428 *** | 0.00385 ** | −0.00258 | 0.00291 ** |
| | (11.35) | (2.32) | (−1.43) | (2.38) |
| LNTG | −0.00145 | −0.0587 ** | −0.225 *** | −0.0415 ** |
| | (−0.14) | (−2.15) | (−6.12) | (−2.11) |
| LNGDP | 0.00447 | −0.00654 | 0.00790 | −0.00812 |
| | (0.85) | (−0.63) | (1.43) | (−1.06) |
| Control region effect | Uncontrolled | Uncontrolled | Controlled | Controlled |
| Control time effect | Uncontrolled | Controlled | Uncontrolled | Controlled |
| AR (1) | 0.021 | 0.033 | 0.070 | 0.026 |
| AR (2) | 0.214 | 0.144 | 0.776 | 0.106 |
| Hansen | 0.192 | 0.681 | 0.221 | 0.263 |
| Sample Size | 240 | 240 | 240 | 240 |

Note: ***, ** and * represent the significance level of 1%, 5% and 10%, respectively.

As shown in column (4) of Table 5, the AR (1) and (2) tests indicate that the residual series of the equations can significantly reject the second-order correlation, although they cannot reject the first-order serial correlation. The autocorrelation test supports that the model is feasible. The corresponding *p* value in the Hansen results is higher than 0.1, which cannot reject the null hypothesis that the instrumental variables are effective, indicating

that the instrumental variables selected in the model are effective; therefor, estimation results of Model IV are consistent and reliable.

In column (4), the coefficient value of the core explanatory variable can be observed. *DFI* is 0.0754, which is significant at the 1% level, indicating that *DFI* has a positive contribution to the agricultural multifunctionality extension. A percentage point increase in *DFI* will result in a 0.0754 percentage point increase in agricultural multifunctionality extension. In addition, the coefficient of agricultural multifunctionality that was lagged by one period ($AV_{it-1}$) is significant at the 1% level, indicating that agricultural multifunctionality itself has typical "path dependence" or "inertia" characteristics, and the future extension of agricultural multifunctionality depends on its past or present. As for other control variables, the coefficients of the urbanization process and industrial structure are negative at 1% and 5% significance levels, respectively, indicating that both the urbanization process and industrial structure change significantly inhibit agricultural multifunctionality extension. The reason may be that both urbanization, characterized by the urban population growth, and industrial restructuring, characterized by the increase in the share of secondary and tertiary industries, have promoted the transfer of surplus rural labor into non-agricultural industries, which has raised the income level of income of farmers and households, while deteriorating the allocation of labor factors to agricultural or rural areas, thereby constraining agricultural multifunctionality extension. Theoretically, urbanization and non-agricultural industry development will promote agricultural multifunctionality extension through the effects of industrial linkage and the multi-demand induction of agricultural products. The significant negative impact coefficient also implies that urbanization and industrial structure adjustment in China has not yet formed a feeder effect on agricultural multifunctionality extension. It is important to strengthen the linkage between urban and rural areas and integrate the three industries, in order to promote agricultural multifunctionality extension and rural revitalization. The coefficient of foreign direct investment is significantly positive at the 5% level in the short term, indicating that the increase in foreign direct investment significantly contributes to agricultural multifunctionality growth. The coefficient of the economic development level is not significant, indicating that there is no intrinsic correlation between the level of agricultural multifunctionality and the regional economic development level, especially in terms of the income level.

### 4.3. Stability Test Based on the Tobit Model

To ensure the credibility and stability of the estimation results, we use the Tobit model for further testing. The Tobit model was proposed by Tobin [38] and its general form is as follows:

$$
\begin{cases}
y^* = \beta X_i + \mu_i \\
y_i = y_i^* & \text{if } y_i^* > 0 \\
y_i = 0 & \text{if } y_i^* > 0
\end{cases}
\tag{3}
$$

In Formula (3), $y_i^*$ is the latent variable, $y_i$ is the observed dependent variable, $\beta$ is the correlation coefficient vector, and the error term $\mu_i$ is independent and subject to normal distribution.

Before using the Tobit model for regression analysis, the selection of a suitable model should be carried out. After passing the Hausman test, the fixed effects model is finally selected. Since the semi-parametric estimation method does not require assumptions about the particular format of the residuals, and consistent estimates can be obtained even in the presence of individual heteroskedasticity [39], this paper f this method to regress the model, and we garnered results for Model III. The coefficient of digital financial inclusion is 0.018, and significantly contributes to the agricultural multifunctionality growth at the 10% level, indicating the robustness of the findings of the benchmark regression.

### 4.4. Heterogeneous Effects of Digital Inclusive Finance on Various Agricultural Segmentation Functions

The previous section verified that digital inclusive finance greatly contributes to agricultural multifunctionality extension, resulting from the evolution of the four functions. Despite being intrinsically related, each segmented function has relatively different evolutionary characteristics. The impact of digital inclusive finance on each agricultural segmented function may also have typical differences, and it may have more policy implications, in order to further discuss the heterogeneous impact of digital inclusive finance on each segmented function on the basis of the overall impact portrayal. Thus, the parameters of Equation (1) were estimated using the System GMM estimation method, with each agricultural segmentation function used as the explanatory variable. The regression results are in Table 6.

**Table 6.** Heterogeneity impact of digital financial inclusion on various agricultural segmentation functions.

| | Agricultural Product Supply Function (1) | Economic Development Function (2) | Social Security Function (3) | Ecological Service Function (4) |
|---|---|---|---|---|
| $AV_{it-1}$ | 0.390 *** | 0.817 *** | 0.874 *** | 0.0750 |
| | (5.15) | (26.60) | (15.62) | (0.40) |
| $LNDIF$ | 0.0248 * | 0.00678 *** | 0.00773 ** | −0.0305 *** |
| | (1.85) | (3.34) | (2.52) | (−5.05) |
| Control variables | Controlled | Controlled | Controlled | Controlled |
| Control region effect | Controlled | Controlled | Controlled | Controlled |
| Control time effect | Controlled | Controlled | Controlled | Controlled |
| AR (1) | 0.017 | 0.015 | 0.082 | 0.020 |
| AR (2) | 0.814 | 0.602 | 0.400 | 0.226 |
| Hansen | 0.283 | 0.253 | 0.210 | 0.571 |
| Sample Size | 240 | 240 | 240 | 240 |

Note: ***, ** and * represent the significance level of 1%, 5% and 10%, respectively.

It can be clearly found that, except for the ecological service function, the lagged periods of the agricultural supply, economic development and social security functions are all significantly positive, indicating that the above three have prominent "path-dependent" characteristics, and that the function level in the previous period will positively influence the function level in the next period. In addition, in terms of significance level, digital inclusive finance promotes the agricultural supply economic development and social security functions at the 10%, 1% and 5% levels. However, as far as the impact coefficient is concerned, the enhancement effect of digital inclusive finance on the agricultural product supply function is more remarkable. It is noteworthy that the coefficient of the effect of digital inclusive finance on the ecological service function is significantly negative, indicating that digital inclusive finance inhibits the ecological service function expansion of agriculture to a greater extent. It may be that the profit- or return-chasing tendency of digital inclusive finance makes it flow towards agricultural product supply, economic development, and social security functions with more obvious economic returns. In contrast, the "public interest" feature of ecological service functions is more evident and the "private returns" of investments are relatively low, which may inhibit the willingness to engage in digital inclusive finance. In addition, because of an improper handling of the relationship between economic and ecological benefits, the excessive pursuit of agricultural supply function, economic development function, and social security function may also cause the former to "squeeze" the latter. Therefore, it is of valuable practical significance to strengthen the green regulation of rural digital inclusive finance, and vigorously develop rural green finance through appropriate environmental regulations to correct the negative aspect of digital inclusive finance on agricultural ecological service functions.

### 4.5. Heterogeneity Test of Each Subdivision Index of Digital Inclusive Finance on Agricultural Multifunctionality

The previous section verifies that digital inclusive finance contributes to improving agricultural multifunctionality. The Digital Financial Inclusion Index does not exist independently, and is subdivided into coverage width, use depth and digitization degree. There may be a greater number of policy implications when examining the heterogeneousness of digital inclusive finance on agricultural multifunctionality extension. The regression results are shown in Table 7 below.

**Table 7.** The heterogeneous impact of each sub-index of digital financial inclusion on agricultural multifunctionality.

|  | (1) | (2) | (3) |
|---|---|---|---|
| $AV_{it-1}$ | 0.644 *** (3.64) | 0.775 *** (10.57) | 0.796 *** (3.78) |
| LNCOV | 0.0599 * (1.80) | | |
| LNUSE | | 0.125 *** (7.19) | |
| LNDIG | | | −0.189 ** (−2.31) |
| Control variables | Controlled | Controlled | Controlled |
| Time effect | Controlled | Controlled | Controlled |
| Rigion effect | Controlled | Controlled | Controlled |
| AR (1) | 0.033 | 0.030 | 0.028 |
| AR (2) | 0.128 | 0.129 | 0.104 |
| Hansen | 0.196 | 0.100 | 0.105 |
| Sample Size | 240 | 240 | 240 |

Note: ***, ** and * represent the significance level of 1%, 5% and 10%, respectively.

As shown in Table 7, the agricultural multifunctionality extension coefficients, lagged by one period, are all positive at the 1% level of significance, showing that the agricultural multifunctionality extension is all significantly path-dependent. The breadth of coverage positively promotes agricultural multifunctionality extension at the 10% level, and depth at the 1% level, but the digitalization degree inhibits agricultural multifunctionality extension at the 5% level. Additionally the digital finance coverage depth index is significantly larger than the breadth, which indicates that agricultural multifunctionality extension is more sensitive to the depth of digital finance coverage. In economic reality, the use depth of digital inclusive finance helps to improve the efficiency of the financial services and create new financial technologies, while reducing the hidden risks caused by uncertainties, such as unstable capital chains. The greater the breadth of digital inclusive finance, the more inclusive it is, highlighting the speed that distinguishes traditional finance from digital finance, and reversing the capital mismatch of traditional finance. This negative correlation of the digitalization degree is caused by the low degree of digitalization of digital finance, which limits agricultural multifunctionality development.

### 4.6. The Regulating Effect of Rural Human Capital

The previous theoretical analysis suggests that the interaction and synergy between rural human capital and digital inclusive finance positively moderate the growth of the development level of agricultural multifunctionality. In this study, the interaction terms of migratory human capital, educational human capital, and health human capital, in the context of digital inclusion finance, are established separately, to further explore the joint effects of rural human capital and digital inclusion finance on agricultural multifunctionality. The estimated results is in Table 8; column (1) shows the interaction effect of rural migratory human capital and digital inclusive finance, column (2) shows the interaction effect of

rural educational human capital and digital inclusive finance, and column (3) shows the interaction effect of rural healthy human capital and digital inclusive finance.

**Table 8.** Interactive effects of rural human capital and digital inclusive finance.

|  | (1) | (2) | (3) |
|---|---|---|---|
| $AV_{it-1}$ | 0.565 *** | 0.603 *** | 0.898 *** |
|  | (6.07) | (4.44) | (7.23) |
| LNDFI | −0.0521 ** | −0.00169 | −0.0467 ** |
|  | (−2.45) | (−0.06) | (−2.24) |
| $LN(DFI \times MH)$ | 0.0201 * |  |  |
|  | (1.81) |  |  |
| $LN(DFI \times EH)$ |  | 0.0128 * |  |
|  |  | (1.89) |  |
| $LN(DFI \times HH)$ |  |  | 0.0268 * |
|  |  |  | (1.77) |
| Control variables | Controlled | Controlled | Controlled |
| Time effect | Controlled | Controlled | Controlled |
| Rigion effect | Controlled | Controlled | Controlled |
| AR (1) | 0.015 | 0.017 | 0.018 |
| AR (2) | 0.215 | 0.141 | 0.166 |
| Hansen | 0.569 | 0.113 | 0.643 |
| Sample Size | 240 | 240 | 240 |

Note: ***, ** and * represent the significance level of 1%, 5% and 10%, respectively.

Columns (1) to (3) show that the estimated coefficient of agricultural multifunctionality development becomes unstable after the three interaction terms of rural human capital and digital financial inclusion are included in the model. This is because after the interaction terms of the three types of rural human capital and digital financial inclusion are added, the impact of digital financial inclusion on the development of agricultural multifunctionality changes from $\beta_1$ in baseline Model I to $\beta_1 + \beta_3 * DFI_{it}$ in Model II. We focus on the interaction coefficient between digital inclusive finance and rural human capital, that is, the synergistic effect, or joint influence mechanism, of different types of rural human capital and digital inclusive finance on the agricultural multifunctionality development.

The coefficients of all three interaction terms between rural human capital and digital inclusive finance are significantly positive, proving that the coordination level improvement between rural human capital and digital inclusive finance will significantly promote agricultural multifunctionality extension. This implies that it is essential to emphasize digital technology-driven, rural, inclusive finance development to alleviate the long-term "financial inhibition" problem faced by agricultural multifunctionality extension, and also to increase the development of rural information, education and training under the macro background of the country, actively promoting the integration of rural industries, the development of agricultural multifunctionality, and the in-depth implementation of rural revitalization. It is also critical to increase the investment in infrastructure and public services, such as rural information, education and training, and medicine and health care, in order to improve the human capital level and comprehensive quality of life of rural residents. Through the coordination of inclusive financial capital and human capital, this can better promote agricultural multifunctionality development.

## 5. Conclusions and Policy Implication

This study tried to construct a comprehensive index system for agricultural multifunctional development, as well as tried to measure the level of agricultural multifunctional development in 30 sample provinces (excluding Hong Kong, Macao, Taiwan and Tibet) in China from 2011 to 2019. We theoretically explain and empirically test the impact of digital inclusive finance on agricultural multifunctional development, and discuss the moderating role of rural human capital in the above impact. The level of agricultural multifunctional development in China continues to increase and reveal large regional differences, with a trend

of higher levels in the east than in the center, and higher levels in the center than in the west. Digital financial inclusion promotes the development of agricultural multifunctionality, but this effect has typical heterogeneity for different segmentation functions. It also promotes agricultural product supply functions, economic development functions and social security functions, but has a significant inhibitory effect on ecological environment functions. Rural migratory human capital, educational human capital and healthy human capital all play a significant, positively moderating role in digital inclusive finance, impacting agricultural multifunctionality development.

To better investigate the role of digital inclusive finance in promoting the level growth of agricultural multifunctional development, we summarize policy implications as follows: first, digital inclusive finance should be incorporated into the policy framework of agricultural multifunctional development, and each region should combine its own agricultural factor endowment conditions with comparative advantages of agricultural multifunctional development. Based on the fundamental principles of sustainable financial development, we should innovate mechanisms and modes of digital inclusive finance to support agricultural multifunctional development, promote the systematic integration and optimal allocation of digital inclusive finance and other agricultural factors, and enhance the comprehensive factor guarantee capacity of agricultural multifunctional development; second, under the guidance of policies and strict environmental regulations, we should guide digital inclusive finance into the green development track. Furthermore, we should establish and improve the service systems available for digital inclusive finance to support agricultural multi-functionality development from multiple aspects/dimensions, such as green credit, green insurance, green direct financing, green financing guarantee and green financial innovation. On the other hand, it is also essential to construct an environmental dynamic monitoring system and environmental information disclosure mechanism of financing projects, using the advantages garnered by digital inclusive finance, in order to better analyze the important role of digital inclusive finance in agricultural ecological services and environmental improvement. Third, we should increase investment in rural communication, education and medical care, improve rural human capital and financial literacy, and promote digital inclusive finance in agricultural multifunctionality developmen.

**Author Contributions:** Conceptualization, Y.W. and Z.T.; methodology, Y.W. and Z.T.; formal analysis, Y.W., Z.T., J.L. and H.H.; resources, Y.W. and Z.T.; writing—original draft preparation, Y.W., Z.T., J.L., H.H. and L.Z.; writing—review and editing, Y.W., Z.T., J.L., H.H. and L.Z.; visualization, Z.T. and J.L.; supervision, Z.T. All authors have read and agreed to the published version of the manuscript.

**Funding:** This work was supported by the Fundamental Research Funds for the Central Universities of Chongqing University (2022CDJSKPT30, 2022CDJSKJC26, 2021CDSKXYGG013).

**Institutional Review Board Statement:** Not applicable.

**Data Availability Statement:** Data supporting reported results available, upon request, from the authors.

**Conflicts of Interest:** The authors declare no conflict of interest.

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
