# Peer review of "Does Digital Inclusive Finance Development Affect the Agricultural Multifunctionality Extension? Evidence from China"

_agriculture, doi:10.3390/agriculture13040804_

Round 1
Reviewer 1 Report
The proposed manuscript addresses the issue of inclusive digital finance that supports the functional development of agriculture. Inclusive digital finance helps to converge income disparities between urban and rural areas and improves the efficiency of agricultural production and operations. This study shows the impact of the digital inclusive finance mechanism on the expansion of agricultural multifunctionality.
The authors consider various functions such as the supply of agricultural products, economic development, social security, and service function separately. Five hypotheses are proposed on the basis of these functions.
1. For data analysis, the article uses an index system of assessing agricultural multifunctionality. The System GMM method is used to reduce the endogeneity of the model and increase the reliability of parameter estimation.
In our opinion, to increase the reliability and accuracy of the research results, it is necessary to use a mathematical apparatus based on neural networks. Neural networks can be trained on individual data samples and produce fairly accurate results. This study does not fully take into account the influence of such parameters as area and time. Therefore, the authors should train the neural network on a certain data sample and then test this model on other regions of China. After that, it is recommended to make a comparative table of the obtained data using the method proposed in the article as well as using a neural network.
Then the readers of the article will clearly see how reliable the data is and what the error of the results obtained is.
The conclusions should describe what economic effect can be obtained based on the data obtained. It is also necessary to describe what further steps should be taken to increase inclusive digital finance for agricultural development.
2. Authors should pay more attention to the design and references to figures and tables in the text. Thus, in paragraph "4.1 The evolution trend" there are references to figure 1, which is absent in the reference list of the manuscript.
3. In figure 2 there is an incomprehensible name of the indicator "National region"!
4. Further on Table 4 presents a consideration of 3 instead of 4 regions shown in Figure 2. The data needs clarification.
Author Response
Reviewer 1
The proposed manuscript addresses the issue of inclusive digital finance that supports the functional development of agriculture. Inclusive digital finance helps to converge income disparities between urban and rural areas and improves the efficiency of agricultural production and operations. This study shows the impact of the digital inclusive finance mechanism on the expansion of agricultural multifunctionality.
The authors consider various functions such as the supply of agricultural products, economic development, social security, and service function separately. Five hypotheses are proposed on the basis of these functions.
1.For data analysis, the article uses an index system of assessing agricultural multifunctionality. The System GMM method is used to reduce the endogeneity of the model and increase the reliability of parameter estimation.
In our opinion, to increase the reliability and accuracy of the research results, it is necessary to use a mathematical apparatus based on neural networks. Neural networks can be trained on individual data samples and produce fairly accurate results. This study does not fully take into account the influence of such parameters as area and time. Therefore, the authors should train the neural network on a certain data sample and then test this model on other regions of China. After that, it is recommended to make a comparative table of the obtained data using the method proposed in the article as well as using a neural network.
Then the readers of the article will clearly see how reliable the data is and what the error of the results obtained is.The conclusions should describe what economic effect can be obtained based on the data obtained. It is also necessary to describe what further steps should be taken to increase inclusive digital finance for agricultural development.
Many thanks for the comments and suggestions. (1) In accordance with the connotation and characteristics of agricultural multifunctionality and with reference to the existing literature, we constructed an agricultural multifunctionality evaluation index system and adopted the entropy weight method to measure the agricultural multifunctionality development index of sample years and sample provinces. For ensuring the scientific and reasonable research, we use the entropy method here mainly based on the following considerations: the objective weights are identified according to the magnitude of the variability of the indicators.The comprehensive evaluation problem is the basic problem of modeling, of which the most commonly used is the entropy method. Entropy method is an objective assignment method, the weights are given by data reaction, not by human ( as hierarchical analysis is a subjective assignment method, with a certain subjective arbitrariness ). The entropy is also called information entropy. The smaller the information entropy, the greater the degree of variation of the index value, the more information it provides, the greater the role it can play in the comprehensive evaluation, and the greater its weight. Conversely, the greater the information entropy of an indicator, the less variation, information, and the role it plays in the comprehensive evaluation, and the less weight it has.
- In empirically testing the effect of digital inclusive finance on agricultural multifunctionality, this study adopts a two-step System GMM estimation method, the comparative advantages of which are explained in the paper.Meanwhile, in the econometric model design and model regression, this study also controls for region fixed effects and time fixed effects, which to some extent exclude the effects of trends that do not vary with region or time and substantially improve the reliability of the model estimation results. In addition, for the robustness of the regression results, we also applied the Tobit model for robustness testing.For the neural network method suggested by the reviewer, our research team does not yet have the specific conditions for the application of this method. Constrained by the conditions, we had to keep the original estimation method in the modification. We still grateful for the expert’s suggestion, which provide valuable approaches for us to use neural network methods in subsequent related studies.
- In the conclusion section, this study reports the overall impact of digital inclusive finance on agricultural multifunctionality and the differences in each function segment, and accordingly proposes policy implications for digital inclusive finance to help expand agricultural multifunctionality.
2.Authors should pay more attention to the design and references to figures and tables in the text. Thus, in paragraph "4.1 The evolution trend" there are references to figure 1, which is absent in the reference list of the manuscript.
Many thanks for the comments and suggestions. Figure 1 was cited incorrectly and we have revised it. The Figure 2 is adjusted to Figure 1.Evolution trend of agricultural multifunctionality extension in China from 2011 to 2019. (Figure 1 of Chapter 4.1 on Page 11)
3.In figure 2 there is an incomprehensible name of the indicator "National region"!
Many thanks for the comments and suggestions.The "National region"(figure 1 in page 11) indicates the spatial and temporal evolution of agricultural multifunctional extension at the national level.(Chapter 4.1 on Page 11)
4.Further on Table 4 presents a consideration of 3 instead of 4 regions shown in Figure 2. The data needs clarification.
Many thanks for the comments and suggestions. We have revised it. In this study, we depict the spatial and temporal evolution trends of agricultural multifunctional extension in four regions: national, central, western, and eastern regions in Figure 1. Accordingly, we have added a description of the mean values of agricultural multifunctionality and segmentation indices at the national level in Table 4 based on the average values of agricultural multifunctionality and segmentation indices in the three regions of East, Central and West. (Chapter 4.1 on Page 12) .

Reviewer 2 Report
The article undertakes a difficult and complex problem to determine the impact of digital financing on multifunctionality of rural areas. The text is long and the empirical analysis is carried out using the right methods. Nevertheless, I have comments and reservations that the authors should take into account when the text is corrected. Below my comments:
1) The essence of digital inclusive finance has not been thoroughly explained. What is their inclusiveness? Are there any digital finances that are not conducive to inclusive inclusion? Which digital finances create financial exclusion and which foster financial inclusion?
2) What is the "financial repression" phenomenon in China's financial supply in agricultural fields?
3) Some of the theses formulated by the authors require better justification, as they cannot be accepted uncritically. For example, thesis was formulated that inclusive digital finance can "provide low-cost, convenient and flexible financial resources to promote agricultural products production and operation activities to maintain or expand reproduction, thus enhancing the supply capacity of agricultural products".
4) In my opinion, the authors formulate very controversial theses on the benefits of digital finance. One could get the impression that they are a panacea for all farmers' problems. However, concrete facts are missing, e.g. what percentage of farmers in the surveyed provinces actually use the online financial platform? How much does a loan obtained in a stationary bank branch cost and how much does a "digital" loan cost? Is the difference big enough to attribute all these "benefits" to financial digital?
5) The selection of some indicators to assess the multifunctionality of agriculture is debatable. For example, what does the variable "Agriculture, forestry and water affairs expenditure per capita" have to do with the multifunctionality of agriculture?
6) Why digital inclusive financial have a negative impact on the ecological service function of agriculture?
7) Digital finance cannot be equated with the benefits of farmers' digital access to information. These are two completely different things.
8) The quality of human capital positively affects the multifunctionality of agriculture, as it requires, among others, creativity and entrepreneurship. The relationship in this respect is direct and there is no need to prove the impact of human capital on the multifunctionality of agriculture through digital finance.
9) Due to the reservations presented above, I think that the conclusions of the research are too far-reaching.
Author Response
Reviewer 2
The article undertakes a difficult and complex problem to determine the impact of digital financing on multifunctionality of rural areas. The text is long and the empirical analysis is carried out using the right methods. Nevertheless, I have comments and reservations that the authors should take into account when the text is corrected. Below my comments:
1.The essence of digital inclusive finance has not been thoroughly explained. What is their inclusiveness? Are there any digital finances that are not conducive to inclusive inclusion? Which digital finances create financial exclusion and which foster financial inclusion?
Many thanks for the comments and suggestions.In the introduction, we have added and improved the connotation and attributes of "inclusive finance" and "digital inclusive finance". The inclusiveness of digital inclusive finance mainly lies in the provision of diversified financial services such as payment, savings and finance for people who are excluded from traditional finance, so that everyone who can access the Internet can obtain financial services at low cost. Therefore, the inclusiveness of digital inclusive finance is relative to that of traditional finance,while "Financial exclusion" is also mainly compared to traditional finance. The main target customers of digital inclusive finance are precisely those who were previously excluded from "financial exclusion", such as small and medium-sized enterprises (SMEs), rural or agricultural people, who are relatively deficient in "collateral". To argue for the inclusiveness of digital inclusion, we have supplemented the service model of digital inclusion for people excluded by traditional finance in the introduction part.
2.What is the "financial repression" phenomenon in China's financial supply in agricultural fields?
Many thanks for the comments and suggestions.In the introduction, we have specifically analyzed the phenomenon of "financial repression" from both supply and demand perspectives. ( paragraphs 2-5 in the introductory chapter on Page 2)
3.Some of the theses formulated by the authors require better justification, as they cannot be accepted uncritically. For example,thesis was formulated that inclusive digital finance can "provide low-cost, convenient and flexible financial resources to promote agricultural products production and operation activities to maintain or expand reproduction, thus enhancing the supply capacity of agricultural products".
Many thanks for the comments and suggestions.We have revised and improved our discussion and added references like Fang and Cai(2022). ( paragraphs 2 of chapter 2.1 on Page 4)
4.In my opinion, the authors formulate very controversial theses on the benefits of digital finance. One could get the impression that they are a panacea for all farmers' problems. However, concrete facts are missing, e.g. what percentage of farmers in the surveyed provinces actually use the online financial platform? How much does a loan obtained in a stationary bank branch cost and how much does a "digital" loan cost? Is the difference big enough to attribute all these "benefits" to financial digital?
Many thanks for the comments and suggestions.This study focuses on the impact of digital inclusive finance on the development of agricultural multifunctionality and conducts an empirical test based on Chinese data. The reviewer's questions are very valuable and innovative,which are worthy of further exploration and help to deepen the understanding of digital inclusion finance.After discussion and consideration, we found that the discussion on these issues requires further investigation with the help of field survey analysis to acquire valuable data and conduct more in-depth research.Therefore, this study still focuses on answering the question of the impact of digital inclusive finance on the development of agricultural multifunctionality.
5.The selection of some indicators to assess the multifunctionality of agriculture is debatable. For example, what does the variable "Agriculture, forestry and water affairs expenditure per capita" have to do with the multifunctionality of agriculture?
Many thanks for the comments and suggestions.At present, the connotation and attributes of agricultural multifunctionality have basically reached a consensus in the academic community, but at the statistical level, there are still some differences understanding especially in terms of the metric index system. Moreover, it is difficult to quantitatively evaluate the level of development of agricultural multifunctionality in the absence of statistical data. In this study, "per capita expenditure on agriculture, forestry and water conservancy" is used as an indicator to characterize the "social security function" of agriculture. The reason is that the increase of this expenditure contributes to the improvement of irrigation conditions in rural areas, and to a certain extent prevents the impact of natural risks such as droughts and floods on agriculture, providing a favorable infrastructure for sustainable agricultural development.
6.Why digital inclusive financial have a negative impact on the ecological service function of agriculture?
Many thanks for the comments and suggestions.The possible reasons for the negative impact of digital inclusive finance on agroecological service functions are explained in 4.4.The possible reason for this is that the profit or return chasing tendency of digital inclusive finance makes digital inclusive finance flow more to agricultural supply, economic development and social security functions with more obvious economic returns, while the "public interest" attribute of ecological service functions is more obvious and the "private returns" of investment " is relatively low, which may reduce the willingness to enter digital inclusive finance.The possible reason for this is that the profit or return chasing tendency of digital inclusive finance makes digital inclusive finance flow more to agricultural supply, economic development and social security functions with more obvious economic returns, while the "public interest" attribute of ecological service functions is more obvious and the "private returns" of investment " is relatively low, which may reduce the willingness to enter digital inclusive finance. Moreover, the excessive pursuit of agricultural supply, economic development and social security functions under the support of digital inclusive finance may also cause the former to "squeeze" the latter due to the improper handling of the relationship between economic and ecological benefits.( paragraphs 2 of chapter 4.4 on Page 15)
7.Digital finance cannot be equated with the benefits of farmers' digital access to information. These are two completely different things.
Many thanks for the comments and suggestions.Digital inclusive finance is more about providing financial support or financing services to farmers for their production and business activities. In the process, it also helps to improve farmers' digital literacy and cognitive ability.
8.The quality of human capital positively affects the multifunctionality of agriculture, as it requires, among others, creativity and entrepreneurship. The relationship in this respect is direct and there is no need to prove the impact of human capital on the multifunctionality of agriculture through digital finance.
Many thanks for the comments and suggestions.The study adds rural human capital investment as a moderating variable and set up an interaction term between digital inclusive finance and rural human capital investment, which helps to recognize better the moderating role played by rural human capital investment in the impact of digital inclusive finance on the development of agricultural multifunctionality.That is, does rural human capital investment further strengthen or weaken "the role of digital inclusion in promoting agricultural multifunctionality". In summary, the introduction of the "interaction term" between digital inclusive finance and rural human capital investment helps to identify the joint impact mechanism of digital inclusive finance and rural human capital investment on agricultural multifunctionality extension.
9.Due to the reservations presented above, I think that the conclusions of the research are too far-reaching.
Many thanks for the comments and suggestions. We have revised and clarified each of the experts' comments.

Round 2
Reviewer 1 Report
The authors did a great job and really improved the quality of the manuscript. I think the paper can be published in the current form